Hidden microalgae diversity in reef systems: reanalysis of coral microbiomes reveals spatial patterns of coral-associated plastid communities in the Southwestern Atlantic Ocean (SWAO)

http://orcid.org/0009-0006-1913-7850 Pires Clara P. 1 clarappires@gmail.com
Villela Livia B. 2
http://orcid.org/0000-0002-5597-6196 Moura Rodrigo L. 2 3
Salomon Paulo S. 1 2 3
http://orcid.org/0000-0002-6030-516X Silva-Lima Arthur W. 1 2 arthurwlima@gmail.com
1 Programa de Pós-Graduação em Biodiversidade e Biologia Evolutiva, Universidade Federal do Rio de Janeiro , Rio de Janeiro, Brazil
2 Departamento de Biologia Marinha, Universidade Federal do Rio de Janeiro , Rio de Janeiro , Brazil
3 SAGE/COPPE, Universidade Federal do Rio de Janeiro , Rio de Janeiro , Brazil
Gottdenker Nicole
Electronic publication date: 2025 Nov 3
Publication date: 2025
Volume: 13
Electronic Location ID: e20116
Received 2024 Jul 18; Accepted 2025 Sep 1
Copyright: © 2025 Pires et al.
Copyright year: 2025
Copyright holder: Pires et al.
License: This is an open access article distributed under the terms of the Creative Commons Attribution License, which permits unrestricted use, distribution, reproduction and adaptation in any medium and for any purpose provided that it is properly attributed. For attribution, the original author(s), title, publication source (PeerJ) and either DOI or URL of the article must be cited.
License URL: https://creativecommons.org/licenses/by/4.0/

Keywords: Metabarcoding, Holobiont, Plastid, Ostreobium, Corallicolid, 16S, Microeukaryote, Protist, Photosymbiont, Symbiont

Funding: Conselho Nacional de Desenvolvimento Científico e Tecnológico (CNPq) through the Programa de Pesquisas Ecológicas de Longa Duração (PELD Abrolhos) Programa de Monitoramento da Biodiversidade Aquática (Fundação RENOVA and FEST/UFES) Programa de Excelência Acadêmica of Coordenação de Aperfeiçoamento de Pessoal de Nível Superior (CAPES PROEX) Fundação de Amparo à Pesquisa do Estado do Rio de Janeiro (FAPERJ) 88887.939820/2024-00, E-26/200.276/2022 CNPq, FAPERJ and CAPES This work was supported by Conselho Nacional de Desenvolvimento Científico e Tecnológico (CNPq) through the Programa de Pesquisas Ecológicas de Longa Duração (PELD Abrolhos) and funded by the Programa de Monitoramento da Biodiversidade Aquática (Fundação RENOVA and FEST/UFES). Clara Paiva Pires was supported through scholarships by Programa de Excelência Acadêmica of Coordenação de Aperfeiçoamento de Pessoal de Nível Superior (CAPES PROEX) and Fundação de Amparo à Pesquisa do Estado do Rio de Janeiro (FAPERJ; Grant Numbers 88887.939820/2024-00 and E-26/200.276/2022, respectively). Rodrigo Leão de Moura and Paulo Sérgio Salomon received long-term support from CNPq, FAPERJ and CAPES. The funders had no role in study design, data collection and analysis, decision to publish, or preparation of the manuscript.

==============================
The microbial community associated with corals plays a critical role in reef ecosystems, yet studies mainly focus on prokaryotes and Symbiodiniaceae, overlooking other oxygen-evolving photosynthetic eukaryotes. This leaves a knowledge gap regarding potentially important microbiome members. Here, we revisited coral microbiome datasets to investigate the diversity of plastid-bearing eukaryotes associated with Southwestern Atlantic Ocean (SWAO) corals. We compiled an inventory of plastid-bearing communities, uncovering their diversity and exploring ecological patterns. We further applied this approach to analyze the plastidiomes (plastid-bearing communities) of corals from the Abrolhos Bank, the largest reef system in the region, as a case study. A systematic literature review of 16S rDNA-based coral microbiome was conducted, excluding studies lacking plastid 16S sequences. The search made in PubMed resulted in 19 studies reporting corals sampled from 2009 to 2022, which were compiled and reanalyzed. Sequences of chloroplast origin (Silva 132) were further taxonomically classified by consensus-BLASTn search with the PR2 plastid 16S database. The dataset encompassed reef water and coral microbiomes from eight coral species, emphasizing the genus Mussismilia. A total of 272 amplicon libraries yielded 707,949 plastid sequences, identifying 196 algal genera across 41 classes. Reef water and coral plastid communities differed markedly. Ostreobium (81%) and Calliarthron (49%) were the most prevalent genera in coral samples. Diatoms were common (>40% of samples) in the water and in corals, whereas corallicolids were exclusive to corals (31.8%). The Abrolhos case study revealed geographic variation in Mussismilia harttii plastidiomes, which were less diverse than those in the water column. Coral indicator taxa included Ostreobium, corallicolids, Navicula, and Amphora. Our findings identify plastidiome variations and their implications for the coral host. Lipid-rich diatoms prevalent in coral plastidiomes may support corals nutritionally after coral bleaching, while other free-living and bloom-forming microalgae produce significantly more reactive oxygen species than Symbiodiniaceae, potentially driving oxidative stress. These results highlight microeukaryotic community variation across corals and its ecological relevance, offering a framework for using plastid-bearing communities as biomarkers of shifts in coral holobionts.

Introduction

Corals are closely associated with a diverse microbial community, including prokaryotes, viruses, fungi, Symbiodiniaceae photosymbionts, and other microeukaryotes; collectively forming the coral microbiome (Ainsworth, Fordyce & Camp, 2017; Peixoto et al., 2017). Most studies have focused on prokaryotes (archaea and bacteria), which significantly influence coral physiology and reef ecosystem (Rohwer et al., 2002; Rosenberg et al., 2007) while also exhibiting temporal and spatial variability in response to biotic and abiotic factors (Sweet, Croquer & Bythell, 2011; Voolstra et al., 2024). In contrast, the diversity and ecological roles of microeukaryotes within the microbiome remain understudied, limiting a comprehensive understanding of the coral holobiont (Keeling & Del Campo, 2017; Bonacolta et al., 2023). Given their broad ecological niches, including pathogens, parasites, heterotrophic and photosynthetic organisms, microeukaryotes may have important but overlooked impacts on coral health (Ainsworth, Fordyce & Camp, 2017; Bonacolta et al., 2024).

Among the photosynthetic microeukaryotes, Symbiodiniaceae (Dinophyceae) are central to reef energy dynamics (Morris et al., 2019). Disruption of this symbiosis during bleaching events is one of the major threats to coral reefs worldwide (Hoegh-Guldberg, 1999), primarily driven by increased reactive oxygen species (ROS) accumulation within the holobiont. In Symbiodiniaceae and other microalgae, ROS mainly arise from electron transfer chains in the photosynthetic reactions, with production increasing under stress conditions such as high temperature and irradiance (Weis, 2008; Foyer, 2018). Despite their central role, Symbiodiniaceae produce relatively low levels of ROS compared to other dinoflagellates and opportunistic bloom-forming algae (Diaz & Plummer, 2018), suggesting that other coral-associated microalgae might contribute substantially to oxidative stress, although their roles remain poorly understood.

Several microeukaryotic groups deserve attention. The chlorophyte Ostreobium is consistently found in the endolithic community within coral skeletons and can contribute to both bioerosion and carbon transfer to the coral host (Fine & Loya, 2002; Tribollet, 2008; Marcelino & Verbruggen, 2016; Ainsworth, Fordyce & Camp, 2017). Diatoms, abundant in the coral mucus layer with densities up to 104 cells/cm2 (Cavada et al., 2011; Costa, Sassi & Gorlach-Lira, 2013), may provide fatty acids that support coral heterotrophy during bleaching events (Houlbrèque & Ferrier-Pagès, 2009). Other less abundant microeukaryotes have been isolated during Symbiodiniaceae culturing efforts, including the pelagophyte Sargassococcus simulans (Varasteh et al., 2022), the chlorophyte Symbiochlorum hainanense (Gong et al., 2018; Aiube et al., 2024), and alveolates such as chromerids and corallicolids whose functions within the coral microbiome are still being elucidated (Moore et al., 2008; Kwong et al., 2019; Keeling, Mathur & Kwong, 2021; Oborník et al., 2012).

Despite their ecological importance, most studies on coral-associated microalgae rely on isolation and culture-dependent approaches. Limited morphological variability among distantly related taxa (Potter et al., 1997) and the high density of Symbiodiniaceae in coral tissues (in the order of 106 cells/cm2; Fitt et al., 2000; Cardoso et al., 2025). Limit the application of microscopic techniques for a comprehensive evaluation of coral-associated microalgae communities. Molecular studies face similar challenges, as universal eukaryotic primers often amplify coral host or Symbiodiniaceae genes preferentially (del Campo, Bass & Keeling, 2020). Strategies to overcome these biases include the validation of a universal set of primers that enhance microeukaryotic detection while minimizing metazoan amplification (del Campo et al., 2019) and taxon-specific primers (Ebenezer, Medlin & Ki, 2012), although the latter restrict broad community characterization.

DNA sequences from organellar genomes are often retrieved in microbiome studies (Sonett et al., 2024). Plastidial 16S rDNA sequences have proven valuable for detecting microeukaryotes in coral samples, such as apicomplexan-related lineages (Janouškovec et al., 2013, 2015; Kwong et al., 2019) and the boring chlorophyte Ostreobium (Verbruggen et al., 2017). However, plastid sequences have not yet been systematically used to assess coral-associated microalgal communities. The development of the curated PR2 plastidial database (formerly PhytoRef, Decelle et al., 2015) allows the analysis of plastid sequences derived from culture-independent studies. Plastid-based community patterns closely resemble those revealed by nuclear markers (18S rDNA; Yeh & Fuhrman, 2022) and correlate positively with picoeukaryote biovolume assessed by optical methods (Pierella Karlusich et al., 2023). Combined metabarcoding approaches using both plastidial (16S and 23S) and nuclear (18S) markers detected chlorophytes and rhodophytes in Australian coral skeletons (Marcelino & Verbruggen, 2016). Nevertheless, a comprehensive analysis of coral-associated plastid-bearing eukaryotic communities remains lacking.

In this study, we revisited coral 16S rDNA microbiome libraries published over a period of more than a decade to inventory plastid-bearing microeukaryotes associated with corals, exploring the prevalence of different microalgae across the Southwestern Atlantic Ocean (SWAO). We further applied this approach to assess geographic variation in the plastid-bearing communities (plastidiomes) of Mussismilia harttii from three reefs in the Abrolhos Bank, the largest reef system in the SWAO (Moura et al., 2013). By combining the reanalysis with a case study, we could explore the prevalence and relative abundance of microalgae in coral-associated plastidiomes and propose a framework for using microeukaryotes as biomarkers of ecological shifts in coral holobionts.

Materials and Methods

Compilation of a coral-associated plastid inventory

A compilation of published 16S rDNA sequence libraries based on culture-independent approaches on environmental coral microbiome diversity was conducted to evaluate plastid community diversity associated with corals off the Brazilian coast in the SWAO (Fig. 1, Fig. S1). A PubMed (https://pubmed.ncbi.nlm.nih.gov/) search was conducted using the keywords: (((microbiome) OR (bacteria*) OR (16S)) AND (Brazil*)) AND ((Mussismilia) OR (coral)). Two reviewers (authors CPP and AWSL) worked together on screening and data collection, with the final search conducted in December 2022. Studies from other oceanographic regions, studies based on culture-dependent methods and reviews were excluded. Studies that met these criteria but did not have sequences available in public databases were excluded from the reanalysis (Fig. 1). Sequences were retrieved from public databases (Table S1) following protocols detailed in each study. Relevant available information associated with these sequences was retrieved, including environmental (e.g., sample location and date, coral host species, health status, coral tissue collected) and technical (e.g., sequencing method, library preparation, 16S primers) metadata (Table S2). The resulting compilation of SWAO coral microbiome articles was complemented by the screening of article citations (de Castro et al., 2013), and an ongoing study from our laboratory (identified as Villela_in_prep in Table S2).

Figure 1 PRISMA 2020 flow diagram for the systematic review.

This flow diagram maps the number of included and excluded studies in different phases of the review. Adapted from PRISMA 2020 guidelines (Page et al., 2021).

Bioinformatic processing and taxonomical identification of plastid sequences

Preprocessing of the retrieved sequences was unique for each sequencing strategy (bacterial cloning, whole-genome shotgun, metabarcoding) due to the specific sequencing artifacts associated with each technique (Fig. S1). Sequences derived from bacterial cloning studies were processed as submitted to the database. Putative ribosomal sequences from whole-genome shotgun studies were filtered with sortmeRNA 4.3.4 (Kopylova, Noé & Touzet, 2012). For metabarcoding studies, cutadapt 1.15 was used to remove primer sequences (Martin, 2011), and denoising was done with the DADA2 algorithm for paired-end sequences (Callahan et al., 2016). For studies targeting the V1-V3 region of the 16S rRNA gene (Zanotti et al., 2020; Zanotti, Gregoracci & Kitahara, 2021; Table S2), paired-end sequences were merged with PEAR v 0.9.11 (Zhang et al., 2014) and denoised with DADA2 as single-end sequences.

Taxonomic classification of the 16S environmental sequence variants (ESVs) was done by Blastn similarity comparisons (Altschul et al., 1990). A sequential classification strategy was employed with representative sequences first compared with the Silva v132 SSU database (Quast et al., 2012). Sequences classified as ‘chloroplast’ or ‘unassigned’ by consensus of the 10 best Blastn results with the Silva database were further reclassified with the PR2 v4.12 plastid database (Guillou et al., 2012; Decelle et al., 2015). This sequential processing ensures that classified sequences originate from plastids, providing taxonomic plastidial classification based on the consensus of the top three BLASTn results against the PR2 database. Sequences unclassified by the PR2 database were excluded. An inventory was compiled using classified plastid sequences and metadata collected from each study (Table S2). Given the unresolved nature of corallicolid taxonomy based on plastid markers (Kwong et al., 2019), we conducted a BLASTn search in the NCBI database with sequences classified as Colpodellida from in situ Abrolhos reef samples, as it was highly abundant in the Sebastião Gomes site. Denoising of the sequencing results and taxonomic classification of the ESVs was done within the qiime2 v 2020.11.1 framework (Bolyen et al., 2019).

The compiled inventory integrates 13 years of microbiome studies over 18 sampling sites in the SWAO. The compilation of results from diverse sequencing methods (with varying sequence lengths, sequencing depths and 16S targets) precludes comparisons of abundance-based metrics, and the prevalence (frequency of occurrence) of the coral-microalgae association was used to assess the consistency of the interaction. Taxonomic identification of plastid ESVs observed in each study is presented at the Genus and Class levels to account for results targeting different regions of the 16S rDNA gene. Water samples were included as a reference environmental matrix to differentiate the coral-associated plastid genera from other genera occurring in reef environments. Plastid genera from in situ studies (177 samples from 17 studies) were ranked based on their prevalence in coral (n = 129) and in water samples (n = 48), and the 15 most prevalent genera in each environmental matrix are presented.

Case study: Mussismilia harttii plastidiome from the Abrolhos Bank

To further illustrate the application of the proposed plastidiome analysis framework, we analyzed Mussismilia harttii microbiomes from an ongoing study in our laboratory. With this case study, diversity and community composition of photosynthetic microeukaryotes are examined in a single sequencing approach, enabling evaluation of microalgae relative abundances. By applying our pipeline to this dataset, we provide an example of how future studies might incorporate our methods for analyzing coral-associated plastidiomes.

Seawater and fragments of visually healthy M. harttii coral specimens were collected from three different reefs—Esquecidos (−18.76710, −39.51788), Parcel dos Abrolhos (−17.99061, −38.67141) and Sebastião Gomes (−17.90654, −39.14580)—throughout 12 months, encompassing two summer seasons (May 2021 and April 2022) and one winter season (September 2021). Sebastião Gomes is a shallow and coastal reef (14 km away from the coast) characterized by higher water turbidity, while Parcel dos Abrolhos is an offshore reef (59 km away from the coast) with clear waters (Teixeira et al., 2021). Among sampling sites, Esquecidos is the deepest reef and is 24 km away from the coast. Samplings were done at different depths for Sebastião Gomes (3 m), Parcel dos Abrolhos (14 m) and Esquecidos (18 m).

Metabarcoding of the ribosomal 16S-V3V4 region was used to characterize the microbiome of the water column (n = 18) and of M. harttii specimens (n = 24). Chloroform-based DNA extractions were done as described in Villela et al. (2024). Amplification of the 16S-V3V4 was done using UPC Hot Start polymerase (Qiagen, Hilden, Germany), with the 55 °C annealing temperature and primers 341F and 785R (Klindworth et al., 2013). Sequencing libraries were prepared with Nextera UD indices and paired-end (2 × 250 bp) sequenced with Illumina technology (50.000 reads per sample), at NGS Soluções Genômicas sequencing facility (Piracicaba, Brazil). Plastid 16S rDNA sequences from this study are published under NCBI BioProject PRJNA1199551. Bioinformatic processing of the sequences was done as described previously, with a sequential taxonomical identification against the Silva v132 SSU database, followed by the PR2 v4.12 plastid database. Samples were rarefied to 987 reads to account for differences in sequencing depth retaining 42 out of 46 samples and ensuring at least four samples per environmental matrix/reef. Rarefaction analysis confirmed that this sampling depth was sufficient to characterize the Shannon diversity of the samples.

To characterize sample alpha diversity, we used the Shannon index (vegan R package; Dixon, 2003). To analyse the differences between matrices, a non-parametric Welch’s test was conducted due to heterogeneity between seawater and M. harttii variances. For each matrix independently (seawater or M. harttii), we conducted an analysis of variance to evaluate the effects of sampling period and reef location on Shannon diversity. Normality of the residuals was verified using Shapiro’s test and homoscedasticity with Levene’s test (Zuur et al., 2009). To further explore significant interactions between these factors, we performed pairwise comparisons using the estimated marginal means (emmeans) approach with post-hoc adjustments for multiple comparisons (Lenth & Lenth, 2018). For each matrix independently, this allowed us to assess differences in Shannon diversity among reef sites within each sampling period. To analyze beta diversity, we applied the robust Aitchison distance (Gloor et al., 2017), followed by MDS analysis to examine dissimilarities between plastid communities. To visualize group clustering and assess significant differences in community composition, we plotted the MDS ordination with 95% confidence ellipses using the stat_ellipse function in ggplot2 (Wickham, 2011). We then used PERMANOVA analysis (p = 999, Oksanen et al., 2013) to assess how sampling reef and period influenced plastid community composition. Finally, to identify plastid genera associated with each environmental matrix and specific sampling reefs in coral samples (Esquecidos, Parcel dos Abrolhos, Sebastião Gomes), we performed an indicator species analysis with the indicspecies package in R (p = 999, Cáceres & Legendre, 2009). All statistical analyses were conducted in R 4.3.0 and RStudio 2024.04.2 (R Core Team, 2023; Posit team, 2024). Scripts used in this study and the results from the taxonomic classification of plastidial sequences are available at https://github.com/arthurwlima/Coral-Plastidiome.

Results

This study provides a comprehensive overview of plastid-bearing eukaryotes associated with scleractinian corals and surrounding reef water in the Southwestern Atlantic Ocean (SWAO). The resulting inventory allowed for quantification of the prevalence of microalgae genera revealing distinct plastid community compositions between coral and reef water samples, as well as spatial variation in coral plastidiomes across reefs of the Abrolhos Bank.

Plastid inventory from Southwestern Atlantic coral reefs

From the 26 studies published between 2009 and 2022, 19 met the criteria for the proposed analysis (Fig. 1, Table S1). A total of 272 samples from 18 locations were re-analyzed, ranging from the Southeast Brazilian coast to the oceanic islands of the Saint Peter and Saint Paul Archipelago, although most samples came from the southern coast of the Bahia state (Fig. 2A). The data set encompassed microbiomes of eight scleractinian coral species, primarily from the endemic genus Mussismilia, and of one hydrocoral species, Millepora alcicornis. Taxonomic classification of plastid sequences identified 196 genera across 41 classes. Genus richness was highest in metabarcoding studies (196 genera), followed by shotgun sequencing (91 genera) and bacterial cloning (28 genera).

Figure 2 Plastid inventory results.

(A) Sampling sites of published in-situ studies in the Southwestern Atlantic Ocean. Pie chart diameters represent the number of samples retrieved from each region, with colors representing coral host genera. (B) The 15 most prevalent genera in water (top) and coral (bottom) microbiomes retrieved from in situ samples according to the PR2 plastid taxonomic classification. The class to which each genus belongs is represented by the bar color. Source of Brazilian coastline: IBGE/DGC. Base Cartográfica Contínua do Brasil, 1:250.000–BC250: 2017 version. Rio de Janeiro, 2017.

Among in-situ samples, Ulvophyceae was the most prevalent class in coral (87.6%, Fig. S2), dominated by the genus Ostreobium, present in 81.4% of coral samples. Other notable Ulvophyceae genera included Bryopsis (19.4%) and Desmochloris, the latter present in 17.8% of coral samples and absent from water samples. Another conspicuous class in corals was the Florideophyceae (75.2%), with Calliarthron being the most prevalent genus (48.8%). Chrompodellids were represented by Colpodellidae, a genus exclusive to corals in our inventory (31.8% of the samples). All nine Colpodellida plastid ESVs showed high similarity (>99% identity) with corallicolid plastid sequences from Kwong et al. (2019) (Table S3). Within the same clade, Vitrella was present in nine Siderastrea stellata tissue samples from Costa dos Corais environmental protection area and one Mussismilia harttii tissue sample from Abrolhos. Among Bacillariophyta, pennate diatoms such as Navicula and Amphora were prevalent in coral samples. Conversely, centric diatoms (Mediophyceae) such as Chaetoceros had a higher prevalence in water samples (39.6%). Other Bacillariophyta genera, such as Cylindrotheca and Thalassiosira, shared a high prevalence in both coral and water samples.

In contrast with the coral plastid inventory, water samples showed a broader taxonomic diversity (Fig. 2B). Over half of the samples had seven distinct algal classes, with Bacillariophyta (79.2%), Prymnesiophyceae (68.8%), and Dictyochophyceae (66.7%) being the most frequent.

Geographic variation in Mussismilia harttii plastid communities from the Abrolhos Bank

A total of 170,173 plastid reads, derived from 42 samples (19 water and 23 coral), were recovered from the Esquecidos (ESQ), Parcel dos Abrolhos (PAB), and Sebastião Gomes (SG) reefs during March, September 2021 and April 2022 (Fig. S3).

Plastidiome diversity was significantly higher in water than in coral samples (Shannon, Welch’s t = −7.3667, p = 0.001, 95% CI = [−1.46 to −0.83]; Fig. 3B, Table S4). Coral plastidiome diversity varied among reef sites (F = 5.226, p = 0.017) but not across sampling periods (F = 0.805, p = 0.464). However, the interaction between site and period was highly significant (F = 9.044, p = 0.002; Table S5). In May 2021, diversity did not differ between ESQ and PAB (estimate = −0.802; p = 0.0544; Table S6). In September 2021, diversity in PAB corals was higher than ESQ (estimate = −1.416; p = 0.0053; Table S6) and SG (estimate = 1.061; p = 0.0244; Table S6), while no significant difference was observed between ESQ and SG (estimate = −0.354; p = 0.6012; Table S6). In contrast, PAB coral plastidiomes had lower diversity than SG in April 2022 (estimate = −1.042; p = 0.0065; Table S6).

Figure 3 Plastid communities in M. harttii tissues and in the water column in the Abrolhos reefs.

(A) Relative abundance of genera found in coral tissue (left) and in water (right) samples in three different sites and sampling dates (ESQ, Esquecidos; PAB, Parcel dos Abrolhos; SG, Sebastião Gomes). Samples from PAB were collected on three different dates, while samples from ESQ and SG were collected on two dates each. (B) Alpha diversity (Shannon) of each reef in coral tissue and water samples across different sampling dates. Statistical significance: *** indicates p-values below 0.001 and ** indicates p-values between 0.001 and 0.01. (C) Multidimensional scaling (MDS) of beta diversity in coral tissue and water samples from each reef reveals clear community dissimilarity between the two matrices, as well as among the different sites.

Plastid communities composition differed significantly by environmental matrix (coral or water, F = 9.65, p = 0.001) and sampling site (F = 3.50, p = 0.001), while the temporal factor alone (F = 1.54, p = 0.027) and its interaction with site (F = 1.55, p = 0.026) have a slight but statistically significant impact (Figs. 3A, 3C, Table S7). Eleven genera were identified as coral plastidiome indicators, including Ostreobium (stat = 0.957, p = 0.001), Colpodellidae (stat = 0.842, p = 0.001), Calliarthron (stat = 0.842, p = 0.001), Bryopsis (stat = 0.5, p = 0.042), Ectocarpus (stat = 0.575, p = 0.019), and two pennate diatoms, Navicula (stat = 0.744, p-value = 0.004) and Amphora (stat = 0.5, p = 0.041) (Table S8). In contrast, 29 genera, including Aureococcus, Teleaulax, and Phaeocystis, were significantly associated with water column plastidiomes (Table S8). Coral plastid communities differed among reef sites (PERMANOVA, F = 5.033, p = 0.001), with no significant effect of time or interaction between time and reef site (Fig. 3A, Table S9). The unresolved genus Corallinales_X.__ (stat = 0.971, p = 0.001) and the genus Chloropicon (stat = 0.691, p = 0.019) were significantly associated with the ESQ reef site (Table S10). The genera Ostreobium (stat = 0.987, p = 0.002) and Colpodellidae (stat = 0.909, p = 0.015) were associated as indicators of both PAB and SG reefs (Table S10).

Water plastidiome diversity was significantly influenced by reef site (F = 22.63, p = 0.001), sampling time (F = 23.21, p = 0.001) and their interaction (F = 52.89, p = 2.28e−06; Table S10). In May 2021, ESQ had higher diversity than PAB (estimate = 0.772, p < 0.0001; Table S12). In September 2021, PAB water plastidiome exhibited significantly higher diversity than ESQ (estimate = −0.353, p = 0.0092; Table S12) and SG (estimate = 0.533, p < 0.0001; Table S12). No significant differences were observed between ESQ and SG (estimate = 0.180, p = 0.1894; Table S12) during this period. No significant differences were observed between PAB and SG in April 2022 (estimate = −0.0225, p = 0.7715; Table S12). PERMANOVA analysis confirmed significant variation in water plastid communities by reef site (F = 3.674, p = 0.001), sampling periods (F = 3.577, p = 0.001), and their interaction (F = 2.753, p = 0.001; Table S13). Indicators for PAB included Phaeomonas (stat = 0.888, p = 0.007), Proteomonas (stat = 0.816, p = 0.016), Rhizosolenia (stat = 0.816, p = 0.022), and Tetraselmis (stat = 0.796, p = 0.019). Chloroparvula (stat = 0.866, p = 0.007) was specific to ESQ reefs, while Cylindrotheca (stat = 0.990, p = 0.001), Vaucheria (stat = 0.985, p = 0.001), and Chloropicon (stat = 0.900, p = 0.041) were associated with SG and ESQ reefs together (Table S14).

Discussion

A community-level perspective on coral-associated, plastid-bearing microeukaryotes

Our plastid inventory, based on reanalyzed 16S rDNA sequences from over 200 samples across 18 sites in the SWAO (Table S2), revealed new insights into the diversity of plastid-bearing microeukaryotes associated with reef corals. Key findings suggest that these organisms are not passively settling from the water column, but instead reflect a structured community likely shaped by selective mechanisms within the coral holobiont. The consistency between the two complementary approaches reinforces our findings: water column indicator genera from Abrolhos matched the 15 most prevalent genera in SWAO water samples and nine of 11 coral indicator genera were also highly prevalent in the coral inventory. Furthermore, spatial variation in plastid-bearing community composition, even within the same host species, points to localized ecological drivers. By integrating a broad-scale reanalysis with a targeted case study in the Abrolhos Bank reefs, we provide a framework for investigating coral-microeukaryotes associations across multiple scales, fostering future research on the potential roles of these microorganisms in coral health and, ultimately, in reef ecosystem dynamics.

Coral plastidiomes are structured by environmental filtering and host selection

The contrasting plastid genera detected in corals vs. water samples was expected and reinforces that plastid-bearing microeukaryotes found in corals are “more than sunken plankton”: not merely passive planktonic cells trapped in the mucus, but rather a selected subset of planktonic taxa shaped by environmental filtering and host selection (Cavada et al., 2011; Moran & Sloan, 2015). Complex molecular cross-talks have been described between coral cells and both bacteria (Franzenburg et al., 2013) and Symbiodiniaceae (Rosset et al., 2021), leading to density regulation, selective digestion, as well as a myriad of cellular mechanisms triggered by host immune responses (Xiang et al., 2020; Wiedenmann et al., 2023; Helgoe et al., 2024). Whether other coral-associated microeukaryotes are regulated by active immune processes or primarily shaped by environmental filtering might depend on the prevalence of their association and the inhabited coral microenvironment.

The recurrent detection of specific microeukaryotes across host species and reef sites in SWAO plastidiomes offers valuable models for exploring coral-microeukaryote interactions. Among the most prevalent, Ostreobium colonizes Pocillopora coral recruits as early as 7 days of larval settlement (Massé et al., 2018), penetrating the skeleton and thriving under low-light and oxygen conditions (Iha et al., 2021). While the impact of Ostreobium on coral physiology remains context-dependent, its ability to transfer photosynthates during bleaching (Fine & Loya, 2002; Ainsworth, Fordyce & Camp, 2017; Tandon et al., 2023) and its ubiquity suggest it may be selectively regulated by the coral host. Diatoms were also prevalent and appear to colonize coral mucus based on their morphology and buoyancy. While centric diatoms tend to remain planktonic, pennate forms like Amphora and Navicula are common in benthic environments (Heil et al., 2004; Gottschalk, Uthicke & Heimann, 2007; Falkowski & Knoll, 2011; Risjani et al., 2021). The high relative abundance of Navicula in the Abrolhos samples reinforces that these pennate diatoms are resident members of the coral mucus, potentially influenced by mucus composition and resource competition within the microbiome (Pisman, Galayda & Loginova, 2005; Thobor et al., 2024). Alternatively, the aggregation behaviour of centric Thalassiosira may explain its prevalence in both water and coral samples (Gärdes et al., 2011).

The high prevalence of diatoms in SWAO corals underscores their nutritional value (Seemann et al., 2013). Following bleaching, when phototrophy-driven energy is scarce, lipid-rich diatoms may provide alternative sources of nitrogen and phosphorus (Ribes, Coma & Gili, 1998; Seemann et al., 2013; Radice et al., 2019; Conti-Jerpe et al., 2020; Wiedenmann et al., 2023). Leal et al. (2014) found that diatom-derived fatty acids are prominent in zooxanthellate corals, highlighting their trophic importance. The predictability of the nutritional resources may support heterotrophy as a coral evolutionary strategy under stress (Hughes & Grottoli, 2013).

Florideophyceae plastid sequences were frequent, spanning multiple coral species and reef sites. Although common in benthic reef environments (Torrano-Silva & Oliveira, 2013), their varying abundance across SWAO reefs suggests environmental modulation. Red algae have also been identified in Pacific coral skeletons (Marcelino & Verbruggen, 2016), and some may possess endolithic life stages similar to the conchocelis phase in Bangiales. Many low-resolution Florideophyceae sequences in our inventory (e.g., “Florideophyceae.__.__.__”) likely correspond to undescribed taxa, as shown in genetic surveys from Canadian waters (Saunders & Brooks, 2023). These findings highlight the need for taxonomic studies of coral-associated Florideophyceae.

Our inventory also revealed less prevalent but ecologically relevant associations, such as Sarcinochrysidaceae pelagophytes (15.5% prevalence in corals) and the alveolate Vitrella (7.6%), with the latter being newly reported in the SWAO. Their occurrence in Symbiodiniaceae culture collections suggests an opportunistic lifestyle and potential for bloom formation in nutrient-rich conditions. Opportunistic pelagophyte blooms have been reported in dead corals on shallow reefs of the Australian Great Barrier Reef (GBR; Schaffelke et al., 2004), and endolithic algae overgrowth has been observed post-bleaching event in the GBR (Fordyce et al., 2021). Such blooms may harm corals via shading, pH shifts, oxygen depletion and ROS or toxin production (Cho et al., 2022; Nguyen & Kim, 2024).

Abrolhos reefs’ coral plastidiomes are spatially structured

Coral plastidiomes showed lower diversity than the surrounding seawater, consistent with host-mediated filtering of microbiota. Similar patterns are seen in coral-associated bacteria (Galand et al., 2023) and Symbiodiniaceae communities (Fujise et al., 2021). Shannon diversity was also influenced by local conditions, as shown by reduced diversity in PAB corals in April 2022, driven by Ostreobium dominance.

Plastidiome composition differed between M. harttii corals and water samples, in agreement with broader trends in the reanalysis. It also varied among reefs, reflecting site-specific environmental factors. Indicator species analysis and MDS results show that PAB and SG reefs shared a similar community compared to ESQ. This may be due to unique ESQ conditions, compared to PAB and SG, which are closer to each other and under higher irradiance levels (Teixeira et al., 2021). In ESQ M. harttii plastidiomes were dominated by red algae, coinciding with visual observations of crustose coralline red algae at the bottom of the polyps (Fig. S3). Although Ostreobium tolerates low light, ESQ may favor rhodophytes such as Corallinales, which use accessory pigments to absorb deep-penetrating light wavelengths (Vásquez-Elizondo & Enríquez, 2017). These results support the hypothesis that Corallinales may either be growing inside the skeleton or have an endolithic phase in their life cycle.

Corals from the shallow, nearshore SG reef face higher anthropogenic pressure and higher thermal stress. Corallicolid sequences were dominant in these samples. SG also had elevated abundances of opportunistic bacteria (Villela et al., 2024), a marker of microbial dysbiosis (Pita et al., 2018). In manipulative experiments, high corallicolid abundance predicted greater coral tissue necrosis after heat stress (Bonacolta et al., 2024). Notably, the corallicolid-dominated microeukaryotic community better predicted thermal stress susceptibility than prokaryotes alone. These results support the hypothesis that microeukaryotes can trigger stress symptoms in the coral holobiont, whereas defining a pathobiome based solely on bacterial taxa remains inconclusive (Pollock et al., 2011; Sweet & Bulling, 2017).

Advantages and limitations of using plastid 16S rDNA sequences

Our results demonstrate the value of plastidial rDNA for characterizing coral-associated photosynthetic microeukaryotes. A key advantage is the reduced amplification of dinoflagellates (including Symbiodiniaceae), whose plastid genomes are degenerated and therefore challenging to detect with 16S primers (Decelle et al., 2015). However, differences in plastid genome structure and copy number across taxa may bias abundance estimates.

We also detected plastid sequences from macroalgae such as Ectocarpus and Calliarthron. Although these may interact with coral physiology, their persistence can obscure signals from microeukaryotes. To minimize such biases, sampling should avoid visible macroalgae and focus on coral tissues and skeletal fragments. Careful sampling improves the reliability of ecological inferences based on plastid-bearing microeukaryotes in coral-associated plastidiomes.

Conclusions

This study uncovered previously underexplored patterns in the community of plastid-bearing eukaryotes associated with corals. The consistent detection of these microeukaryotes across the SWAO suggests they serve as valuable models for investigating how corals regulate their microbiomes. Our case study from the Abrolhos reefs further demonstrated spatial variability in coral plastidiomes, indicating that these communities respond to local environmental conditions.

We show that coral microbiomes can be expanded to include 16S rDNA sequences from bacteria and plastids without increasing sampling or sequencing efforts. By characterizing associations beyond Symbiodiniaceae and prokaryotes, this approach fosters a more integrative understanding of the coral holobiont. Incorporating protists into coral-microbiome models may reveal key inter-phylum interactions and disease processes, offering a broader framework for coral resilience and guiding conservation strategies.

Supplemental Information

Supplemental Information 1 Flowchart illustrating the analytical framework for studying plastid-bearing eukaryotes in coral microbiomes.

The workflow starts with a bibliographic review to compile coral-associated 16S rRNA sequence datasets from the Southwestern Atlantic Ocean (SAO), including sequences derived from bacterial cloning, shotgun, and metabarcoding studies. Preprocessing for the retrieved sequences and other bioinformatic processes are illustrated in hexagons. Classified plastid sequences and their associated metadata are compiled into the SAO plastid inventory (Table S2), enabling further ecological analyses. Two main analysis were conducted: the prevalence assessment of plastid-bearing eukaryotes with in-situ samples from the inventory and a reanaylisis of Mussismilia harttii plastidiomes from Abrolhos. Ecological analyses are illustrated in circles.

Supplemental Information 2 Most prevalent classes observed in in situ samples of the plastid 16S inventory, separated by the environmental matrix (coral or water).

Supplemental Information 3 Characterization of Abrolhos Marine Park reefs in Bahia, Brazil.

The top panel illustrates the locations of the Esquecidos Reef (ESQ), Sebastião Gomes Reef (SG), and Parcel dos Abrolhos Reef (PAB) within the Abrolhos Marine Park in Brazil. Coral reefs are marked in blue, submerged reefs are indicated, and the boundaries of the Abrolhos Marine Park are outlined in red. The inset shows the location of the Abrolhos Marine Park within Brazil. The bottom panel shows the benthic community and structure found in each reef. Photo credit: Rodrigo Leão de Moura. Source of Brazilian coastline: IBGE/DGC. Base Cartográfica Contínua do Brasil, 1:250.000 – BC250: 2017 version . Rio de Janeiro, 2017.

Supplemental Information 4 Metadata on studies used in the compilation of the plastid 16S rRNA sequence inventory.

The "study_id" column was used to reference the work in the plastid inventory.

Supplemental Information 5 Number of occurrences of the genera observed in the inventory of plastid 16S sequences, and associated metadata.

The first 196 columns concern PR2 taxonomic classification at genus level and following columns are metadata related. On column “host_species” coral species abbreviations are: Mussismilia braziliensis (MBR), Mussismilia hispida (MHIS), Mussismilia harttii (MHAR), Tubastrea tagugensis (T_TAG), Siderastrea stellata (SST), Mussismilia (MUS), Porites astreoides (PO_AS) and Millepora alcicornis (MI_AL). Columns “study_id” can be used to reference plastid data on Table S1, “index” has accession numbers for NCBI and “Library_Name” are the original sample names from each study.

Supplemental Information 6 Colpodellida plastid ESVs BLAST with the NCBI database.

Supplemental Information 7 Statistical tests results used in Abrolhos reefs.

Supplemental Information 8 PRISMA checklist.

Supplemental Information 9 The rationale for conducting the systematic review on plastidial coral microbiome.

Supplemental Information 10 PRISMA 2020 abstract checklist for the systematic review. Adapted from PRISMA 2020 guidelines (Page et al., 2021).

Special thanks to the Marine Phytoplankton Lab team at UFRJ for their invaluable guidance and feedback throughout this research. We acknowledge the use of the free version of ChatGPT (OpenAI) for English grammar revision. We sincerely thank the reviewers for their insightful comments and suggestions, which greatly contributed to the maturation of this article. Lastly, we would like to acknowledge the efforts of our fellow researchers committed to reproducible science, whose publicly available data contributed to the work herein.

Additional Information and Declarations

Competing Interests

The authors declare that they have no competing interests.

Author Contributions

Clara P. Pires conceived and designed the experiments, performed the experiments, analyzed the data, prepared figures and/or tables, and approved the final draft.

Livia B. Villela performed the experiments, prepared figures and/or tables, and approved the final draft.

Rodrigo L. Moura performed the experiments, authored or reviewed drafts of the article, and approved the final draft.

Paulo S. Salomon conceived and designed the experiments, authored or reviewed drafts of the article, and approved the final draft.

Arthur W. Silva-Lima conceived and designed the experiments, performed the experiments, analyzed the data, prepared figures and/or tables, authored or reviewed drafts of the article, and approved the final draft.

Data Availability

The following information was supplied regarding data availability:

Code is available at GitHub and Zenodo:

- https://github.com/arthurwlima/Coral-Plastidiome.

- Arthur Lima, & Clara Pires. (2025). arthurwlima/Coral-Plastidiome: Southwestern Atlantic Ocean corals (v_1.0). Zenodo. https://doi.org/10.5281/zenodo.14833738.

Data is available in the Supplemental Files.

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
