# Peer review of "Hidden microalgae diversity in reef systems: reanalysis of coral microbiomes reveals spatial patterns of coral-associated plastid communities in the Southwestern Atlantic Ocean (SWAO)"

_PeerJ, doi:10.7717/peerj.20116_

## Round 0.1 · original submission · Major Revisions

· Academic Editor

Major Revisions

Your manuscript has been reviewed, with one reviewer suggesting rejection and the other suggesting major revision. I strongly recommend responding to each suggestion and re-submit.

Reviewer 1 ·

Basic reporting

The manuscript aims to inventory plastid-bearing eukaryotes in corals from the Southwestern Atlantic Ocean. However, it employs three different approaches to investigate the plastid-bearing eukaryote communities in coral and water samples without providing a clear rationale for the selection of these methods.

While the literature references appear appropriate and the figures are generally clear, the manuscript's structure is confusing, making it difficult to follow the figures. For instance, Figure S1 is crucial for understanding the main text but is presented as a supplementary figure.

Unfortunately, the manuscript lacks clarity and specificity in its methods and results, and the discussion is overly speculative. Additionally, the introduction is lengthy, poorly structured, and fails to present a clear hypothesis.

Experimental design

The manuscript employs three distinct approaches that do not interact with each other, making it difficult to identify a central research question.

The methods section lacks clarity, particularly in the subsection "Geographic Variation in Mussismilia harttii Plastid Communities from the Abrolhos Reef Bank," which is based on unpublished data.

Validity of the findings

Unfortunately, the research has the potential for significant novelty and impact, particularly given the limited exploration of plastid-bearing eukaryotes associated with corals in the Southwestern Atlantic Ocean. The study reveals intriguing findings, such as the high prevalence of Colpodellidea, yet these results are not fully explored or adequately discussed in the manuscript. Overall, the conclusions drawn are speculative and lack depth.

Additional comments

The manuscript could be divided into three separate papers, each focusing on a specific approach. This would allow for clearer methods, more detailed results, and a more in-depth discussion for each aspect of the research.

Reviewer 2 ·

Basic reporting

The authors are also missing an acknowledgments section and have inconsistent formatting throughout the MS, sometimes with 1-2 spaces in between paragraphs, sometimes with none. I strongly suggest closer editing prior to your next submission.

Experimental design

In this study, the authors conducted a systematic literature review of coral microbiome studies as well as an experiment examining the effects of pollution on a coral’s microbiome. While I commend the authors for combining a substantial re-analysis of the existing literature with an experimental case study, as written, the paper lacks flow and clear motivation. In particular, the authors should draw explicit links between their review with their case study, as it was not clear 1) why the review was necessary for the experiment examining the effects of pollution on corals and 2) how this information was used to inform the analyses conducted on Mussismilia hartii, and 3) why this method allowed for novel insights on environmental pollution (i.e. what would have been lost if they had used “standard” methods?). The authors are also missing an acknowledgments section and have inconsistent formatting throughout the MS, sometimes with 1-2 spaces in between paragraphs, sometimes with none. I strongly suggest closer editing prior to your next submission. I do believe this is a study that is worth publishing, but believe it requires substantial revisions before doing so. Specific comments are as follows:

Validity of the findings

Introduction:
In general, the introduction is very lengthy and lacks flow. It also fails to introduce the experiment on Mussismilia hartii and the questions surrounding environmental pollution – why was this coral used? What hypotheses did you have? For general writing, I strongly suggest you check to make sure each paragraph has a clear topic sentence. As of now, the intro mostly reads as a free-flowing list of facts about corals and various microeukaryotes.

Lines 79-82: Can be combined.

Lines 84-87: Seems irrelevant to this study, so can be deleted.

Lines 92-94: Unclear relevance.

Lines 96-136: There is far too much detail for the introduction on specific microbes, as it is never clear how these taxa fall into the central questions of the paper (i.e. what can be gained from focusing on microeukaryotes as opposed to bacterial members of of the coral holobiont). Suggest drastically cutting this section down to a single paragraph.

Lines 138-145: Unclear relevance – can be cut.

Lines 147–158: This is too much detail for the intro and you still have yet to introduce your focal coral or your questions regarding environmental pollution.

Methods:
As noted in the general notes on the introduction, some general background on the study site, organisms, and pollutant is desperately needed. Also, please note which packages you’re using for your analyses (e.g. permanova was conducted using the vegan package (Oksanen et al. 2022)).

Line 223: In general, some detail on the study site and coral would be useful in this section.

Lines 256-259: I am not familiar with this micropollutant. Please provide some background on this and why it was an appropriate treatment for this site and coral.

Line 263: It would be helpful noting earlier that EE2 is a hormone. Are these concentrations based on previously published studies?

Line 284: Did you include any error correction? Running multiple post-hoc GLMs for individual groups of eukaryotes seems unwise and is likely unnecessary given that you’re already running a PERMANOVA.

Results:
I would strongly suggest providing readers an indication of effect size/directionality of trends, especially in the linear models. Which sites/samples had high/low diversity, greater abundance of XYZ microbes, etc.? Also, do not say, “ABC stat test indicated that...” Just tell us the trend and cite the test in the parentheses along with the relevant statistical information. I am also concerned about the prevalence of algae in your coral holobiont samples, as this could indicate sample contamination.

Lines 299-301: Should be in methods.

Lines 314-315: Should be in methods.

320: The _XX in Bryopsidales_XX can probably be removed, as this is a common class of algae. I am surprised this and Corallinales_X later one were kept in the coral samples, as these are common species in the epilithic algal matrix and CCAs, respectively, and could indicate sample contamination with unwanted members of the benthos.

Lines 349-350: This is a sentence fragment.

Line 350: What test stat are you referrring to in these comparisons?

Line 378: remove “the analysis of variance revealed that” and move the stat information to the parenthetical directing the reader to the appropriate figure. Also, give us information on directionality – was it greater or lower in the coral samples? Effect size?

Line 397: Remove “permanova analysis identified”

Discussion:
Much of the issues I had with the introduction are also present in the discussion, which similarly lacks flow and fails to connect the key ideas/goals of the experiment. I am a big believer in starting your discussion with a paragraph to remind the readers of the context and a brief summary of your key results to highlight how your study moved the field forward. I think this would greatly improve the flow of the paper before you jump into the nitty gritty details.

Lines 418-419: How does this indicate anything about health and resilience? This point needs to be expanded on further.

Lines 419-424: Again, I’m not sure how the abundance of these taxa indicates anything about their use for corals.

Lines 425-427: This needs to be expanded on further – how does this complement or improve upon the more commonly used methods?

Lines 432-435: Repetitive to results – can cut.

Lines 438-440: See Franzenburg et al. 2013 PNAS

Lines 440-442: This sentence is redundant – can be cut.

Lines 440-469: This is a lot of information but I’m unclear on what purpose it serves – what are you trying to show by highlighting these trends? What would be the topic sentences for these paragraphs? This is what should drive your discussion of certain groups.

Lines 417-479: This is a spurious conclusion – prevalence of algal sequences in a coral sample does not indicate the sample is coming from a macroalgal-dominated reef, nor does it provide any information about competition for resources. It indicates potential issues with sampling (which you acknowledge).

Lines 481–488: Again, not clear what this is contributing to the story you’re trying to tell.

Lines 501–512: Much of this is repetitive to results and can be cut.

Lines 507-512: Perhaps I misread things, but this again just sounds like contamination of coral samples.

Lines 514-521: Again, you can’t draw any conclusions about algal abundance based on the methods and results you’ve presented here, nor can you draw any conclusions about reef/coral health. This seems very likely to have just been sample contamination, and likely should be dropped.

Line 532: In general, I’m missing some way to connect this to broader trends in the ecology of this system. What was the point of the pollution experiment? What are the implications of these results?

Lines 534-535: Repetitive to results. Also, does this indicate that microbiomes are constantly changing? Or that corals needed to be acclimated longer? This needs further interpretation.

Lines 538-548: Missing some interpretation here – are these changes good or bad for corals? What are the hypothesized function of these taxa based on the literature?

Lines 550-560: This sounds like it could be a fatal issue in your experimental design – especially given that most of your trends were driven by time. Could the microbiome shifts have been largely dictated by your mesocosm setup as opposed to your manipulation?

Lines 565-575: Again, missing some of the “so what” here. Was the purpose only to show that you can also look at microeukaryotes? Or do they provide an important, separate piece of the story compared to more commonly used methods?

Lines 581-589: You really need to tie in the field and mesocosm here.

Line 593: Missing section

Figures:

Figure 2: Missing x axis labels

Figure 3: Figure 3a should have cleaner x-axis labels. I would appreciate connecting letters reports in figure 3b to indicate significantly different groupings (See emmeans package for how to do this). I also think figure 3C would benefit from ellipses noting 95% confidence intervals or significant groupings.

Figure 4: Figure 4a is missing x axis labels. Figure 4c would benefit from ellipses OR a line/arrow showing the path of microbial communities through the mean centroid of each treatment over time (i.e. a successional trajectory). Tetraselmis should be italicized in figure 4d y axis.

Additional comments

N/A

Reviewer 3 ·

Basic reporting

• Language and Clarity: The manuscript is not clearly written, and the arguments lack clarity. The title suggests a focus on corals, but the abstract and main text do not align with that focus, which causes confusion. The research objectives and conclusions are not well communicated.
• Literature and Background: The authors did not provide sufficient background context or literature references. The research gap is not clearly identified.
• Structure and Data Sharing: The manuscript lacks clear structure, and raw data, as well as codes or databases, are not made available for reproducibility. A proper acknowledgment section is also missing.

Experimental design

• Research Question: Although the research is relevant, the authors do not explicitly define how their work fills an identified knowledge gap. The introduction and motivation for the study are unclear.
• Technical and Ethical Standards: The technical rigor is adequate, but improvements are needed. For example, the reviewer suggests using mock metagenomes to validate the method.
• Methodology: The methods section lacks clarity and sufficient detail for replication. The reviewer suggests including a flowchart of the analytical approach, and also encourages better organization of the results.

Validity of the findings

• Impact and Novelty: The study lacks a clear focus on its impact and novelty. The discussion should be better structured to highlight the potential implications of the findings.
• Data Robustness: The data and statistical analyses are insufficient. The authors should validate their methods using mock data to ensure robustness.
• Conclusions: The conclusions are not adequately tied to the original research question, and clearer biological insights are needed.

Additional comments

In the manuscript titled “Hidden microalgae diversity in reef systems: reanalysis of coral microbiomes reveals temporal and spatial patterns of coral-associated plastid communities in the Southwestern Atlantic Ocean (SAO),” Pires et al. present a novel and relevant analytical approach for quantifying micro-eukaryotic sequences in coral metagenomes. By exploring this under-studied diversity, the work has important implications for coral ecology. However, to fully harness the potential of this research, I encourage the authors to consider some modifications to enhance clarity, strengthen the methodology, and improve the impact of the findings.

The current structure of the manuscript could be improved, as the line of argumentation is not always clear, and the objectives are somewhat ambiguous. To address this, I suggest restructuring the manuscript to place greater emphasis on the validation of the method. Since the key contribution of the paper lies in the proposed approach, validating it with mock metagenomes (for each metagenome type used) would provide robustness to the results. This validation would help demonstrate that the method accurately detects microeukaryotic diversity, lending more credibility to the findings. Including performance metrics such as sensitivity, specificity, and accuracy would further highlight the strength of the approach.

I also recommend reordering the presentation of the results. A visual flowchart illustrating the analytical approach should be introduced early in the manuscript. This figure could outline the sample types, the key steps in bioinformatics analyses, and how the method was applied to both case studies. After this, the validated results from the mock samples should be presented, followed by the results of the two case studies. This approach would provide a logical and clear progression, making the paper easier to follow while showcasing the relevance and precision of the methodology.

Furthermore, focusing the discussion more clearly on the ecosystem services provided by photosynthetic eukaryotes—such as coral nutrition—would emphasize the biological relevance of the work. The methods section could outline key groups of interest (based on existing literature) to monitor in coral-associated plastid communities, and the results could discuss any significant changes observed in these groups. Framing the study this way would allow for more meaningful biological conclusions, demonstrating the broader significance of the proposed method and findings.

I also suggest the authors add an acknowledgments section. This important section would provide context for how the work was made possible, including any funding sources and contributors who supported but did not author the paper. Additionally, the authors should make the scripts, data, and reference databases publicly available in a repository (e.g., GitHub) to ensure reproducibility, which is crucial for a study proposing an analytical approach.

By incorporating these changes, the manuscript could be greatly strengthened, making it a more influential contribution to the field. This work has the potential to be widely adopted, and refining the presentation and validation will help ensure it reaches its full impact.

Abstract

The abstract needs to be carefully revised. This is a very important section of the manuscript. There is no clear logical flow where the authors…

The title focuses on corals, but in the abstract, the authors reveal that “free-living” samples and a microcosm experiment were analyzed. It is not clear to me what the central question/objective of the work is. Reading the abstract…

The main focus of the work is to describe—could the authors perhaps replace this word with “explore” or “investigate,” already bringing in their objectives the practical consequences of this new knowledge?

Generally, “Coral microbiome studies focus on the prokaryotic community and the Symbiodiniaceae family” focuses on the microbial community (Bacteria and Archaea). The importance/introduction of the problem could and should be placed more clearly and accurately at the beginning of the abstract.

I suggest putting the objective first—the authors extensively describe what they did in methodological terms, but the motivation for conducting this study is unclear. How does this work contribute to advancing knowledge?

The authors state as a goal: “Our aim is to compile an inventory of the plastid communities retrieved from SAO corals and analyze ecological patterns within these communities using 16S metabarcoding.” The authors could better formulate their objectives, expressing more clearly the relevance of the presented study, as it is important.

The authors describe the methods excessively, even listing the names of tools/databases: “The search conducted in PubMed resulted in 19 studies reporting corals sampled from 2009 to 2022, which were compiled and reanalyzed. Sequences identified as of chloroplast origin (Silva 132) were further taxonomically classified by consensus-blastn search with the PR2 plastid 16S database.” I question whether this level of detail is relevant for this section.

I propose the following reflection for the authors: How do they want the work to be cited? The main results should be clear and connected so that the article can eventually be cited just by reading the abstract.

Following the logic of the previous comment, I suggest that the authors better organize the flow of information, making better connections between the sentences.

It is unclear what the findings of the work are. It is not clear what the implications of the work are by reading the abstract. What is conveyed is that the work is highly focused on developing an approach/methodology to evaluate the diversity of photosynthetic eukaryotes, but it was not formulated to describe this approach, rather the result of the data analysis from it.

Introduction

The introduction does not present the study in a logical, clear, and objective sequence. The authors present a series of facts in a way that does not clearly convey the motivation for conducting the study or its relevance. I suggest revising the entire introduction, organizing the facts so the reader can understand the importance and objective of the study. The authors often jump between facts related to corals and then discuss the reef environment. In the collection of public data, the authors use data from manipulative experiments. There is no clear hypothesis for this, no sequence of questions or goals. Lastly, I suggest that at the end of the last paragraph, the authors provide a preview of the results, without focusing exclusively on the methods. What is the main finding of the paper? From reading the abstract and introduction, this was not clear to me.
Some specific points below.

Line 87 – Be cautious when using the term “coral-microbial interactions.” Most microbiome studies only show the presence/abundance of taxa found in coral tissues. I suggest caution, as interactions (e.g., parasitism, commensalism, mutualism) involve more than just presence, and measuring/evaluating these interaction mechanisms goes beyond microbial abundance. I suggest the authors focus on a metric that is relevant on its own, such as community structure. This does not diminish the work done, and the term is more accurate for the present manuscript.

Line 98 – Is there no more recent reference with more precise measurements of coral reef production? From 1984 to the present day, many primary producer groups have been studied from entirely different perspectives. The cited reference is a classic, but given what the authors propose (a new approach), it might make sense to use newer methods and ways to evaluate this information.

Lines 138-145: I found this line of reasoning very interesting. The authors could center their argument on this, focusing on coral nutrition and other ecosystem services provided by photosynthetic eukaryotes, to support the effort. In the methods section, the authors could suggest important groups (based on evidence from the literature) and in the results, report changes in these groups for the case studies.

Line 149 – Check spelling and style throughout the text. In this line, the authors write, “We will utilize plastid 16S gene sequences for taxonomic identification.” As far as I understand, the authors used these sequences, so they should report it in the past tense.

Methods

I suggest always using “to do/test this, we did that.” I think the most important figure for the methods would be a flowchart illustrating the innovative process the authors are proposing in the present study, i.e., “Compilation of a coral-associated plastid inventory,” instead of the modified PRISMA workflow.

It is unclear why coral and seawater samples were used in the study, as all the arguments in the introduction point to coral sample analysis. In the methods, the keywords do not mention water either.

I question how the sections “Geographic variation on Mussismilia harttii plastid communities from the Abrolhos Reef Bank” and “Environmental response of Mussismilia harttii plastid communities in a mesocosm experiment” were proposed. I can imagine why the authors chose to include these two analyses, but readers should clearly understand the rationale behind these choices. As far as I understand, the work begins by proposing a method/approach to explore/identify the diversity of photosynthetic eukaryotes using culture-independent methods (or simply through DNA sequencing – perhaps this is a more appropriate way to put it). Then, the authors apply the proposed method to answer specific questions with original data, validating their method. If my understanding is correct, I suggest that the authors clarify this, explicitly stating their hypotheses. This understanding is based on the statement in line 255 (“Another study used as proof of concept for the proposed plastid community investigation method”).

I illustrate the lack of clarity in explaining objectives by pointing out that the authors present information very descriptively.
Lines 256-259: “A microcosm experiment was conducted to evaluate the bacterial microbiome in visually healthy Mussismilia harttii collected from Santa Cruz Cabrália, Coroa Vermelha Reef, Brazil, in response to the persistent micropollutant ethinylestradiol (EE2).” Was this study conducted to evaluate the microbial community’s response to pollutant exposure? Changes in which metrics? Diversity, structure, composition? What did this study find in terms of microbial groups? Was there a significant effect? This would clarify the need to investigate changes in the photosynthetic eukaryote community. At the beginning of the method description, the authors could clarify that these results are published and provide a brief introduction to the main findings.

Lines 273-288: I suggest that the authors present descriptions of all bioinformatic and statistical analyses in separate sections (i.e., one for bioinformatics and another for statistics), explicitly detailing what was done in each case.

Results

I suggest the authors provide a brief initial description of the results, a paragraph outlining the efforts and main findings of the work, which will be further elaborated in specific sections.

I suggest restructuring the results section to reflect the authors’ efforts. First, they structured a method for recovering the diversity of photosynthetic eukaryotes, followed by case studies with appropriate conclusions. I question whether it would be essential to validate the methods, perhaps using mock samples, showing that the approach indeed works. I also question the effort to provide a general description of the corals (and water) of the SAO, as the results of the first case study clearly show that the environment significantly affects the composition of this specific community (line 346). It would be interesting to know if the microbiota (Bacteria and Archaea) follows similar patterns.

Section: “Geographic variation on Mussismilia harttii plastid communities from the Abrolhos Reef Bank”
Without understanding what the treatments/locations mean, reading the reported results makes little sense, and the reading becomes monotonous. Since there is no explanation of the experimental/sample design, and the motivation for using this case study to test the proposed approach is not clearly stated, it is very difficult for the reader to follow and build an understanding of what is being reported.

Section: “Response of Mussismilia harttii plastid communities to estrogen application”
This is the first time the word estrogen is mentioned. As previously stated, it would be important for the authors to include these “details” at the beginning of the manuscript.

Discussion

Overall, the discussion lists a series of findings without illustrating their significance. If the suggestions I have made above are helpful in improving the manuscript, I suggest the discussion be completely restructured, following the example set by the authors in the paragraph in lines 444-465, where they explain possible ecological roles. Note that in this paragraph, the authors mentioned the lifestyles of diatoms, which have expected behaviors. If the authors reported the results from this perspective, the manuscript would be clearer and, I imagine, more impactful (as I suggested at the beginning of the document).

Line 532: Avoid using abbreviations in section titles.

---

## Round 0.2 · Major Revisions

· Academic Editor

Major Revisions

Reviewer 2 has highlighted that this revision is much improved, and I agree. However, they point out a need to improve flow and improve linkages with literature review to the case study.

Reviewer 2 ·

Basic reporting

The writing is generally improved throughout and all previous formatting issued were addressed. I still think the authors should draw clearer and more explicit connections between their lit review and their case study throughout the manuscript.

Experimental design

The removal of the pollution case study has made the paper significantly more focused.

Validity of the findings

See Additional comments below for detailed notes on this.

Additional comments

The manuscript is generally improved over the previous version, especially the methods and results sections. However, it still lacks a clear flow and needs to explicitly link the results of the literature review to the case study. There are interesting results that will be valuable to the broader scientific community, but the MS still needs more revisions prior to publication. Specific comments are as follows:

Abstract:
The abstract contains all the relevant pieces that make this study interesting, but should be restructured to present this information more clearly. Specifically, I would introduce the case study at the end of the first paragraph. E.g. Lines 49-52 could read, “We first compiled an inventory of plastid communities [....] We then tested this approach by analyzing the plastidome of corals from the Abrolhos Reef Bank[...]” Or something along those lines.
Lines 72–79: This should be a single paragraph.
Introduction:
The introduction still lacks cohesion and does not adequately set up both aspects of the experiment. I also don’t think the subheadings are necessary. Rather, they highlight the general lack of flow as the authors are moving through the various concepts. Some suggested changes are outlined below.
Lines 84-87: You can cut the definition of what a coral reef is and just start by focusing on corals as holobionts.
Lines 103-120: This paragraph can be cut down substantially – the point seems to be highlighting that eukaryotes are also important in the bleaching response. Start with this, as it will help transition from the previous paragraph.
Lines 122-127: Unclear relevance – either cut entirely or cut down substantially with clearer connection to prior passages. If kept, it should not stand alone as a paragraph.
Lines 129-142: This also is too much detail for what is effectively just an illustrative example of what roles microeukaryotes might play. Thus, this whole section can be cut down substantially and combined into a single paragraph, thus flowing directly from the opening paragraph.
Lines 160-172: This paragraph is missing something to lead directly into the next one – it should set up the final knowledge gap that this paper will fill.
Lines 176-179: Change “Additionally, we applied” to “We then applied” to strengthen the connection between the two parts. Then, add a brief sentence or clause explaining why these analyses were a necessary and logical complement to your review.
Methods:
Generally much improved. My only major suggestion is to specify how you checked model assumptions in lines 273-283.
Results:
Also generally improved. The figures are much better and clearer in this draft and help illustrate the main trends that are discussed in the text.
Discussion:
Generally speaking, the discussion is lengthy and would benefit from a revision enhancing clarity and flow. Most paragraphs do not have a clear topic sentence and read as a slightly more detailed summary of results without a clear throughline or narrative. This is most apparent from lines 404-488. It’s often unclear how the results compared to the authors’ hypotheses or contribute to the broader literature. Rather than compile a list of bullet points about the most common organisms observed in your samples, consider how the presence of each complements each other and contributes to our broader understanding of the system.
Lines 386–395: Include a sentence or a clause here explicitly linking your review to your case study as part of your summary of the paper. How did each part elevate the other? This will help orient the reader as they move through the rest of the discussion.
Line 399: Was this contrary to expectations?
Lines 399-403: See also Franzenburg et al. 2013 PNAS for a mechanism by which cnidarians might regulate their associated species.
Lines 404-410: What are the implications here? Hypothesized function? Unclear why this is important or relevant.
Lines 413-416: Can be cut.
Lines 417-422: Somewhat speculative. Can be shortened substantially.
Lines 423-434: Another paragraph that needs a clear topic sentence. It seems that the main takeaway here is that differences in eukaryotes between samples are due to their buoyancy, but it is unclear why this is important for understanding the role of the plastidome.
Lines 441-443: Interesting, but again – how does this tie directly to your results, and how do your results further the understanding of this phenomenon?
Lines 473-488: Lengthy and repetitive to earlier.
Line 510: Vibrio should be italicized
Lines 515-517: Unclear phrasing here – generally speaking, I would focus first on the eukaryotes, then the prokaryotes.
Lines 522-541: Again, you need to make the connections between the lit review and the case study more explicit. This sort of just comes out of nowhere here after lengthy discussions of the results from your case study.
Lines 548-550: I’m not sure if your results do show this, or at least I have not been convinced of this from the previous text. This could be remedied during a substantial restructuring of the discussion section.

---

## Round 0.3 · Minor Revisions

· Academic Editor

Minor Revisions

Thank you for the revisions. Reviewer 2 commented that most revisions were adequately addressed-and has some comments for minor revisions.

Reviewer 2 ·

Basic reporting

The manuscript is improved throughout. No further comments here.

Experimental design

The design and methodology is as clear as it's ever been. No further comments.

Validity of the findings

Again, much improved. No further comments.

Additional comments

Overall I commend the authors for their substantial revisions in improving this manuscript. I found a few minor typos that can be quickly and easily addressed, but the MS is in great shape and should be good to publish shortly.
Specifics below:

Abstract
Overall much improved. My only suggestions is to revise slightly to make it clear when you’re transitioning from discussing the lit review results to the case study results in the main body of the abstract text.

Introduction
Much improved. Reads well and flows cleanly between paragraphs.

Methods
Line 229 – correct “analysis of variance” to welch’s test

Results
Overall much improved

Discussion
Much improved and I commend the authors for their work here.

Lines 375-377: Comma splice with the central clause – unclear subject verb agreement

Line 414: Missing space after period following “environmental factors.”

---

## Round 0.4 · Minor Revisions

· Academic Editor

Minor Revisions

Reviewer 1 has had only 1 minor revision and as soon as these are addressed, the manuscript is ready for acceptance. All reviewers thus far have indicated the value of this paper to the literature in this field.

Reviewer 1 ·

Basic reporting

The manuscript "Hidden microalgae diversity in reef systems: reanalysis of coral microbiomes reveals spatial patterns of coral-associated plastid communities in the Southwestern Atlantic Ocean (SWAO)" has been significantly improved from the first version I read. I recommend the manuscript for publication after addressing the very minor reviews.

Line 84 - Dinophyceae (not Dynophyceae). This was the only misspelled name I found; however, a careful review of all taxonomic names is recommended.

Experimental design

The experimental design is clear and, in general, sufficiently described. However, I think a better explanation about how the decision to rarify the number of reads (line 223-224) should be supplied. Also, could you please provide the Illumina sequencing machine and DNA library prep kit that were used (line 219-220)?

Validity of the findings

The data is well presented, including the supplementary results, and is robust.

Reviewer 2 ·

Basic reporting

As noted in the last review, I feel this paper is ready for publication save for a few very minor changes, which the authors addressed.

Experimental design

As noted in the last review, I feel this paper is ready for publication save for a few very minor changes, which the authors addressed.

Validity of the findings

As noted in the last review, I feel this paper is ready for publication save for a few very minor changes, which the authors addressed.

Additional comments

As noted in the last review, I feel this paper is ready for publication save for a few very minor changes, which the authors addressed.

---

## Round 0.5 · accepted · Accept

· Academic Editor

Accept

This manuscript is ready for publication. All reviewers' minor comments have been adequately addressed.